# Exercise-induced increase in blood-based brain-derived neurotrophic factor (BDNF) in people with multiple sclerosis: A systematic review and meta-analysis of exercise intervention trials

Parnian Shobeiri[1,2,3,4], Amirali Karimi[1], Sara Momtazmanesh[1,2,4], Antônio L. Teixeira[5], Charlotte E. Teunissen[6], Erwin E. H. van Wegen[7], Mark A. Hirsch[8], Mir Saeed Yekaninejad[9]*, Nima Rezaei[4,10,11]*

1 School of Medicine, Children's Medical Center Hospital, Tehran University of Medical Sciences (TUMS), Tehran, Iran, 2 Systematic Review and Meta-Analysis Expert Group (SRMEG), Universal Scientific Education and Research Network (USERN), Tehran, Iran, 3 Non–Communicable Diseases Research Center, Endocrinology and Metabolism Population Sciences Institute, Tehran University of Medical Sciences, Tehran, Iran, 4 Research Center for Immunodeficiencies, Pediatrics Center of Excellence, Children's Medical Center, Tehran University of Medical Sciences, Tehran, Iran, 5 Department of Psychiatry and Behavioral Sciences, Neuropsychiatry Program, McGovern Medical School, The University of Texas Health Science Center at Houston, Houston, Texas, United States of America, 6 Department of Clinical Chemistry, Neurochemistry Laboratory, Amsterdam Neuroscience, Amsterdam University Medical Centers, Vrije Universiteit, Boelelaan, Amsterdam, The Netherlands, 7 Department of Rehabilitation Medicine, Amsterdam Movement Sciences, Amsterdam Neuroscience, Amsterdam UMC, Location VUmc, Vrije Universiteit Amsterdam, Amsterdam, The Netherlands, 8 Department of Physical Medicine and Rehabilitation, Carolinas Medical Center, Carolinas Rehabilitation, Charlotte, North Carolina, United States of America, 9 Department of Epidemiology and Biostatistics, School of Public Health, Tehran University of Medical Sciences, Tehran, Iran, 10 Department of Immunology, School of Medicine, Tehran University of Medical Sciences, Tehran, Iran, 11 Network of Immunity in Infection, Malignancy and Autoimmunity (NIIMA), Universal Scientific Education and Research Network (USERN), Tehran, Iran

☙ These authors contributed equally to this work.
* yekaninejad@yahoo.com (MSY); rezaei_nima@yahoo.com (NR)

## Abstract

### Background

Exercise training may affect the blood levels of brain-derived neurotrophic factor (BDNF), but meta-analyses have not yet been performed comparing pre- and post-intervention BDNF concentrations in patients with multiple sclerosis (PwMS).

### Objective

To perform a meta-analysis to study the influence of exercise on BDNF levels and define components that modulate them across clinical trials of exercise training in adults living with multiple sclerosis (MS).

**Data Availability Statement:** All relevant data are within the manuscript and its Supporting information files.

**Funding:** The authors received no specific funding for this work.

**Competing interests:** The authors have no competing interests to declare.

**Abbreviations:** AD, Alzheimer's disease; BDNF, brain-derived neurotrophic factor; CI, confidence interval; CNS, central nervous system; CNTF, ciliary neurotrophic factor; EDSS, expanded disability status scale; ELISA, enzyme-linked immunoassay; GDNF, glial cell line-derived neurotrophic factor; IGF, insulin-like Growth Factor; IQR, interquartile range; MCI, mild cognitive impairment; MMP-2, -9, matrix metalloproteinase-2, -9; NGF, nerve growth factor; NT-3, -4, neurotrophin-3, -4; PD, Parkinson's disease; PPMS, primary-progressive multiple sclerosis; PwMS, patients with multiple sclerosis; RRMS, relapsing-remitting multiple sclerosis; S1P, sphingosine-1-phosphate; SD, standard deviation; SMD, standardized mean difference; SOCS1,3, suppressor of cytokine signaling-1, -3; SPMS, secondary-progressive multiple sclerosis; TNF-α, tumor necrosis factor α; VDBP, vitamin D-binding protein.

## Method

Five databases (PubMed, EMBASE, Cochrane Library, PEDro database, CINAHL) were searched up to June 2021. According to the Preferred Reporting Items for Systematic Reviews and Meta-Analyses, we included 13 articles in the meta-analysis, including 271 subjects. To investigate sources of heterogeneity, subgroup analysis, meta-regression, and sensitivity analysis were conducted. We performed the meta-analysis to compare pre- and post-exercise peripheral levels of BDNF in PwMS.

## Results

Post-exercise concentrations of serum BDNF were significantly higher than pre-intervention levels (Standardized Mean Difference (SMD): 0.33, 95% CI: [0.04; 0.61], *p-value* = 0.02). Meta-regression indicated that the quality of the included studies based on the PEDro assessment tool might be a source of heterogeneity, while no significant effect was found for chronological age and disease severity according to the expanded disability status scale.

## Conclusion

This systematic review and meta-analysis shows that physical activity increases peripheral levels of BDNF in PwMS. More research on the effect of different modes of exercise on BDNF levels in PwMS is warranted.

## 1. Introduction

Brain-derived neurotrophic factor (BDNF) is a member of the neurotrophin family, which plays an essential role in neuroregeneration and neuroprotection [1]. More specifically, BDNF participates in various fundamental domains as follows: (1) neuronal and oligodendroglial survival and growth, (2) neurotransmitter modulation, and (3) neural plasticity. BDNF modulates several brain functions, such as memory and learning, by having a major role in the development of brain circuits [2]. Besides the central nervous system, BDNF has been found in different organs, such as the brain, lungs, heart, spleen, gastrointestinal tract, and liver. Actually, several cell types, including neurons, glia cells, fibroblasts, vascular smooth muscle cells, and thymic stroma release BDNF [2].

Reports on the alterations of the BDNF levels in patients with multiple sclerosis (PwMS) are controversial, but overall, BDNF is usually increased during relapse, with normal levels during remission phases [3–5]. When the demyelination process is actively evolving, BDNF levels tend to increase, probably as a compensatory mechanism against neuronal and glial damage [6]. In addition to microglia, astrocytes and neurons, central nervous system (CNS) infiltrating Immune cells (e.g., T cells and phagocyte cells) can secrete BDNF in demyelinating MS lesions [6].

Supervised exercise or physical activity has been a promising nonpharmacological therapeutic approach for PwMS to help them with their functional capacity and mental health [7]. A systematic review of the literature showed that five out of eight studies demonstrated positive impact of exercise on cognitive status of PwMS [8]. Furthermore, several studies have reported a lower annual relapse rate in PwMS who do exercise compared to patients without regular physical practice [9]. Conversely, several studies have shown significant effects of

exercise on increasing peripheral BDNF concentrations in healthy populations in parallel with improved quality of life and well-being [10–13], which indicates a potential neuroprotective effect for exercise. Furthermore, several studies have reported a lower annual relapse rate in MS patients who underwent exercise compared to patients without regular physical practice [9].

Considering that axonal loss and cerebral atrophy occur in MS, exercise prescription could promote neuroprotection, neuroregeneration, and neuroplasticity and reduce long-term disability by increasing BDNF levels. In this context, exercise-induced BDNF increase has been regarded as one of the underlying mechanisms supporting the positive effects of exercise in PwMS [14]. Physical activity increases DNA demethylation in the BDNF promoter region, resulting in higher production of the neurogenesis-promoting signaling molecule [15]. Increased TrkB receptor (a BDNF receptor in the astrocytes) sensitivity may be linked with increased BDNF sensitivity, necessitating the reduction of BDNF synthesis and release following exercise [16]. Notably, prolonged physical activity raised BDNF mRNA levels in the hippocampal dentate gyrus, which was associated with substantial changes in the amounts of TrkB [17].

As exercise training is becoming a vital component in the therapeutic toolbox of MS patients, to date, international guidelines are available [18]. Rehabilitation groups, e.g., International Progressive MS Alliance, have suggested that future research should focus on progressive MS rehabilitation and investigate the effect of exercise on the pattern of disease progression in these patients [19]. It is worth mentioning that pioneers in the field developed a framework to strengthen the standardization, quality and scope of MS rehabilitation studies that would help PwMS with their quality of life (MoXFo initiative) [20].

Herein, we aimed to conduct an up-to-date meta-analysis to determine whether exercise training significantly affects peripheral BDNF concentrations in PwMS. We also discuss the association of these markers with chronological age, EDSS score, and the quality of the existing literature.

## 2. Materials and methods

This review was registered (#CRD42021256621) in the International Prospective Register of Systematic Reviews (PROSPERO). The Preferred Reporting Items for Systematic Reviews and Meta-Analyses (PRISMA 2020) guidelines [21] were followed for this meta-analysis.

### 2.1. Search strategy

We performed an online search in PubMed, EMBASE, PEDro database, CINAHL, and Cochrane Central Register of Controlled Trials (Cochrane Library) databases up to June 1st, 2021, to detect original studies with focus on BDNF changes after exercise in MS patients. Medical Subject Headings (MeSH) and Emtree were used to retrieve PubMed and Embase results, respectively. Our search strategy is presented in the supplementary material. In addition, reference lists of retrieved studies were searched for additional relevant reports.

### 2.2. Selection criteria

Studies were included if (1) they were peer-reviewed clinical trial articles, (2) BDNF blood levels were measured quantitatively using enzyme-linked immunoassays (ELISA) or other assays, (3) BDNF measured before and after an exercise intervention, and (4) the exact values of the BDNF marker were either given within the manuscript or provided by the authors of the original study for performing the meta-analysis. Exclusion criteria were as follows: (1) pediatric MS, (2) case reports, case series, letters, commentaries, abstracts, protocols, review articles,

and animal and in vitro studies. Two authors (P.S and A.K) independently performed the screening and eligibility assessment. In case of discrepancy, the two authors discussed and consulted with the third author (S.M) and resolved the conflict.

## 2.3. Data extraction

Two reviewers independently extracted (1) bibliographic information (study title, year of publication, first author, study type, and country), (2) demographic and clinical features of the sample (number of patients and controls, age, sex, disease duration, mean expanded disability status scale [EDSS] score), (3) methodological details (diagnostic criteria, characteristics of the ELISA or other assay), and (4) levels of the BDNF before and after the intervention. We communicated with the studies' corresponding authors for additional information if the absolute values of the levels of BDNF were not given in the published manuscript. The inter-rater reliability between reviewers was calculated using the kappa coefficient [22].

## 2.4. Quality assessment

The methodological quality of the included studies was assessed by two reviewers (P.S. and S. M.) independently, based on the PEDro scale [23]. PEDro is a reliable and valid checklist consisting of 11 items as follows: (1) eligibility criteria, (2) random allocation, (3) concealed allocation, (4) baseline comparability, (5) masked participants, (6) masked therapists, (7) masked assessors, (8) adequate follow up, (9) intention to treat analysis, (10) between-group comparison, and (11) point estimates and variability [23–25]. Note that the eligibility criteria item does not contribute to the total score. Thus, each study gains a score from 0 to 10. We categorized studies based on their PEDro score; below 4 as "poor" quality, a score between 4 and 5 indicating "fair" quality, a score of 6 to 8 considered to be of "good" quality, and a score of 9 to 10 indicating "excellent" quality [26]. Any disagreements were resolved by discussion between two reviewers.

## 2.5. Statistical methods

We estimated a standardized mean difference (SMD) (Hedges' g) and 95% confidence interval (CI) for each between-group comparison as the included studies were conducted in a 16-year period and were susceptible to having different ELISA assays. The SMD of $\leq 0.2$, 0.2–0.8, and $\geq 0.8$ represented small, moderate, and large effect sizes, respectively. Meta-analyses were performed for comparisons for which results from at least three individual datasets were available.

If the values reported in the manuscript were given as a median and interquartile range (IQR) or median and range, and we were not able to retrieve the mean ± standard deviation (SD) from the authors, we used statistical methods suggested by Luo et al. [27] and Wan et al. [28] to convert these values.

To assess heterogeneity between studies in the between-group meta-analyses, we used Cochrane's Q-test and the $I^2$-index. The $I^2$-indices of $\leq 25\%$, 26–75%, and 75%$\leq$ represented low, moderate, and high heterogeneity degrees, respectively [29]. We utilized random effect models according to the DerSimonian and Laird method [30]. Random-effects models are preferred if significant heterogeneity is expected, as they account for variable underlying effects in estimates of uncertainty, including both within- and between-study variance. We visualized the results of the meta-analysis as forest plots and the drapery plot. The drapery plot is a supporting figure to forest plot and was proposed to demonstrate confidence intervals assuming a fixed significance threshold and prevent researchers from exclusively relying on the *p-value* < 0.05 significance threshold [31,32].

To further assess the causes of heterogeneity, we conducted a sensitivity analysis to identify influential cases for meta-analyses with significant heterogeneity and including ten or more studies. Each time we omitted one study and recalculated the effect size (Leave-One-Out Analyses). To reduce the heterogeneity among individual studies, we conducted a subgroup analysis based on the type of intervention used in each study.

Publication bias was initially assessed by visual observation of the degree of funnel plot asymmetry. Then, we used Egger's bias test [33] and Begg-Mazumdar Kendall's [34] to objectively confirm the visual perception from the funnel plot. A *p-value* < 0.1 was considered as evidence of publication bias. Funnel plots and Egger's plots are available. When there was evidence of publication bias, we adjusted the effect sizes using the trim-and-fill method [35].

All computations and visualizations were carried out using R version 4.0.4 (R Core Team [2020]. R: A language and environment for statistical computing. R Foundation for Statistical Computing, Vienna, Austria), and STATA 16 (StataCorp. 2019. Stata Statistical Software: Release 16. College Station, TX: StataCorp LLC) for metaregression and Egger's plots. We used following packages: "meta" (version 4.17–0), "metafor" (version 2.4–0), "dmetar" (version 0.0–9), and "tidyverse" (version 1.3.0). All forest plots and the drapery plot were designed using R. A *p-value* of <0.05 was considered statistically significant.

## 3. Results

### 3.1. Selection of studies

The search strategy retrieved a total yield of 199 studies. After the removal of duplicates, 153 studies remained. Title/abstracts screening identified 32 potentially eligible studies, and 13 original clinical trials met the criteria to be included in the meta-analysis [36–48]. No further studies that were appropriate for inclusion were identified via hand searching and checking references. Fig 1 illustrates the process of study selection according to the PRISMA guideline.

The agreement between the two independent reviewers for study selection was excellent for both titles/abstracts (kappa = 1.00; percentage agreement = 99.98%) and full-text (kappa = 1.00; percentage agreement = 100%).

### 3.2. Study characteristics

Studies were published between 2004 and 2021, and a total of 271 patients with MS participated in the 13 studies included in the current analysis [36–39,41–48]. Outcome measures in these studies were serum BDNF [36–42,44–48], plasma BDNF [43], IL-6 [36,37,46–48], nerve growth factor (NGF) [36,37,40], matrix metalloproteinase-2,-9 (MMP-2, MMP-9) [42], neurotrophin-3,-4 (NT-3, NT-4) [45], glial cell line-derived neurotrophic factor (GDNF) and ciliary neurotrophic factor (CNTF) [45], insulin-like Growth Factor (IGF) [44], sphingosine-1-phosphate (S1P) [43], suppressor of cytokine signaling-1,-3 (SOCS1, SOCS3) [41], tumor necrosis factor α (TNF- α) [37], vitamin D-binding protein (VDBP) [40], and serotonin [42]. Given the type of exercise in each study, we categorized them into three main groups; (1) six studies used aerobic programs [36,37,42,46–48], (2) five studies applied combined training [38,40,41,44,45], and (3) two studies stated anaerobic training [39,43]. In addition, the frequency of exercise of included studies varied from two to three sessions per week, and intervention varied from 6 to 60 min per session over a period of three to 24 weeks.

All but two studies measured BDNF through enzyme-linked immunosorbent assay (ELISA). Bansi et al. [37] and Wens et al. [38] used flow cytometry and Meso Scale, respectively, for BDNF assessment. The characteristics of included studies are detailed in Table 1.

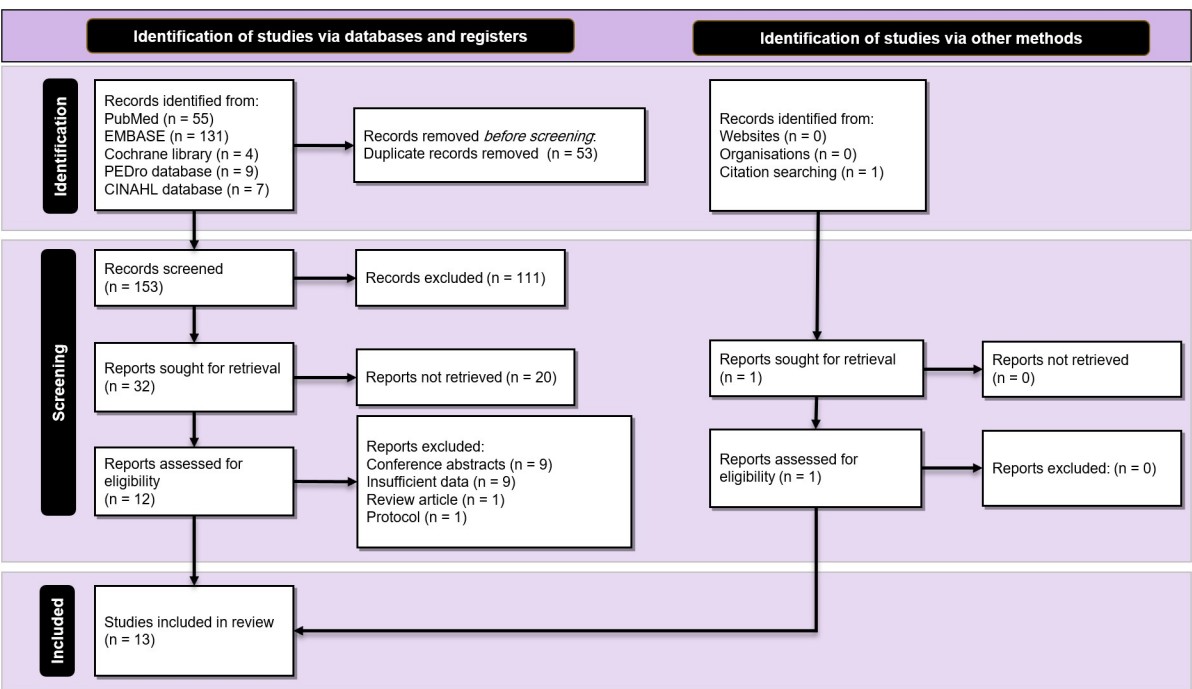

**Fig 1. PRISMA diagram.** Study selection process according to the Preferred Reporting Items for Systematic Reviews and Meta- Analyses (PRISMA) guideline.

### 3.3. Participants' characteristics

The combined mean ± SD of the age of participants was 43.72 ± 8.45 years for 271 reported patients. The cumulative number of female and male participants was 153 and 51, respectively, reported by ten studies [36–38,40–43,46–48]. Nine studies reported the EDSS score of the patients [36–38,40–44,48], and the combined EDSS mean ± SD was 3.40 ± 1.5 for 189 reported patients. The combined mean ± SD of the disease duration time of the participants was 9.5 ± 8.7 years, reported by four studies [41,42,44,48]. Not all studies reported the type of MS, but there were 122 RRMS [38,41–43,45,48], 31 SPMS (secondary-progressive multiple sclerosis) [42,46,47], and 6 PPMS (primary-progressive multiple sclerosis) [46,47] patients. The study by Khademosharie et al. [40] did not indicate the number of patients in each MS group (SPMS or PPMS).

### 3.4. Quality assessment

The median total PEDro score was 6 (IQR = 2.5; mean ± SD = 6.3 ± 1.3; range: 4 to 9) of 10, which indicated an overall good quality of the included studies (Table 2). All studies met the following criteria: baseline comparability, between-group statistical comparison. None of the included studies met the criteria of clinician rater and participant blinding. The agreement between the two reviewers for assessing study quality was excellent (kappa = 1.00; percentage agreement = 100%). Due to the nature of the interventions, in none of the included studies were the participants or therapists blinded.

### 3.5. Comparison of pre- and post-intervention BDNF concentration

Baseline serum levels of BDNF were significantly higher after exercise intervention (SMD 0.3309, 95% CI [0.0434; 0.6148], *p-value* = 0.0275, test of heterogeneity: $I^2$ = 51.5%, *p-value* =

**Table 1. Baseline and exercise intervention characteristics of included studies.**

| Author, Year | Country | Exercise protocol | Session's settings | Outcome measures | BDNF measurement protocol | N (Type of MS) | Diagnostic criteria | Age (mean ± SD), years | EDSS Score (mean ± SD) | Disease duration (mean ± SD), years | Female/Male | Pre-exercise BDNF levels (mean ± SD), pg/ml | Post-exercise BDNF levels (mean ± SD), pg/ml |
|---|---|---|---|---|---|---|---|---|---|---|---|---|---|
| Schulz et al., 2004 | Germany | An 8-week (2x/week) bicycle ergometry training program tailored to their individual capabilities (aerobic training) | 30 min at 75% of maximum power in watts | BDNF, IL-6, sIL-6R, NGF | ELISA (Promega, Madison, WI, USA) Serum | 13 | Poser Criteria | 39 ± 9 | 2 ± 1.4 | NA | 9/4 | 4353 ± 3217 | 5930 ± 5178 |
| Bansi et al., 2013 | Switzerland | A 3-week (2x/week) cardiopulmonary exercise test (aerobic training) | 3 minutes at rest (no pedalling) on the cycle ergometer; 3 minutes of unloaded pedalling as a warm up; testing phase until the participant reached a symptom limited maximum | BDNF, IL-6, sIL-6R, NGF; TNF-α | Flow cytometry (Zug, Switzerland) Serum | 52 | revised McDonald criteria | 50.84 ± 7.84 | 4.65 ± 0.93 | NA | 35/18 | 17154.2615 ± 8431.0054 | 18815.7538 ± 7956.1332 |
| Wens et al., 2016 | Belgium | A 24-week (five sessions per 2 weeks) progressive combined training program (Combined training) | Continuous aerobic exercise, 1x6min (start)- 3x10min (final); intensity 12–14 RPE + 35 min resistance exercise; volume: 1x 10 rep—4x15 rep; intensity 12–14 RPE. | BDNF | Meso Scale (Discovery, Rockville, MD, USA) Serum | 15 (RRMS) | McDonald criteria | 42 ± 3 | 2.7 ± 0.3 | NA | 9/6 | 11092 ± 1005 | 12020 ± 942 |
| Ozkul et al., 2018 | Turkey | An 8-week (2x/week) combined exercise training consisting of aerobic training and Pilates training (Combined training) | 25 min aerobic exercise; intensity 60–70% HR$_{max}$ (week 0–4), 70–80% HR$_{max}$ (week 5–8), + 50 min resistance exercise with elastic bands; gradual intensity; 10 reps (week 0–4), 20 reps (week 5–8). | BDNF, SOCS1, SOCS3 | ELISA (Shanghai Sunred Biological technology, Shanghai, China) Serum | 18 (RRMS) | McDonald criteria | 34.59 ± 13.88 | 1.36 ± 1.01 | 6.17 ± 6.84 | 14/4 | 1663.2309 ± 2480.1192 | 1947.6375 ± 2553.4374 |
| Eftekhari et al., 2018 | Iran | An 8-week Mat Pilates training | 40–50 min of pilates training; Sets: 1–2 Reps: 3–10; Rest: 60 sec between sets. | BDNF, IL-10 | ELISA (Boster Biological Technology Ltd) Serum | 13 | McDonald criteria | 34.46 ± 7.29 | NA | NA | NA | 10678850 ± 2260170 | 11550140 ± 2619600 |
| Khademosharie et al., 2018 | Iran | A 12 week (3x/week) (combined training) | (2 resistance/1 aerobic) Resistance: Sets: 2–4, Intensity: 60–80% RM; Reps: 8–14; Rest between series 2–3 min Endurance: 2–4 sets with 4–13 reps at 40–55% HRR; Rest between sets (3–4 min). | BDNF, NGF, VDBP | ELISA (Boster Biollogical Technology, Pleasanton, CA, USA) Serum | 24 (PPMS/SPMS) | revised McDonald criteria | 36.76 ± 6.87 | 3.1 ± 0.5 | NA | 24 | 4223 ± 2084 | 4707 ± 1918 |
| Zimmer et al., 2018 | Germany | A 3-week cardiopulmonary exercise test (aerobic training) | 30 minutes of training, with warm up and cool-down for the first and last 2 minutes. | BDNF, Serotonin, MMP-2, MMP-9 | ELISA (R&D Systems, Inc., Minneapolis, MN, USA) Serum | 27 (14 RRMS, 13 SPMS) | revised McDonald criteria | 51 ± 9.9 | 4.37 ± 1.39 | 11.98 ± 11.34 | 20/7 | 7526.9 ± 10606 | 24663 ± 13019 |

(*Continued*)

**Table 1.** (Continued)

| | Study ID | | | | | | Patients | | | | | | Results | |
|---|---|---|---|---|---|---|---|---|---|---|---|---|---|---|
| Author, Year | Country | Exercise protocol | Session's settings | Outcome measures | BDNF measurement protocol | N (Type of MS) | Diagnostic criteria | Age (mean ± SD), years | EDSS Score (mean ± SD) | Disease duration (mean ± SD), years | Female/ Male | Pre-exercise BDNF levels (mean ± SD), pg/ml | Post-exercise BDNF levels (mean ± SD), pg/ml |
| Jørgensen et al., 2019 | Denmark | A 24-week (2x/week) resistance training (resistance training) | Sets: 3–5, Intensity: 10 repetitions at 15-RM and 6 repetitions at 6-RM; Rest between sets: 2–3 min; Lower limbs: 4 series of 10 repetitions at 10-RM | BDNF, S1P | ELISA (Promega, Madison, WI, USA) Plasma | 16 (RRMS) | NA | 45.09 ± 8.94 | 3 ± 0.81 | NA | 12/4 | 168838.1 ± 154842.5 | 153309.9 ± 109674 |
| Abbaspoor et al., 2020 | Iran | An 8-week (3x/week) aerobic and resistance training (Combined training) | 15–20 min continuous aerobic exercise, intensity at 55–70% HRmáx + 35 min resistance exercise; volume: 1 set (week 1–4)—2 sets (week 5–8); intensity 10–13 (week 1–4), 13–16 (week 5–8) RPE. | BDNF, IGF-1 | ELISA (Shanghai crystal day biotech, China) Serum | 10 | NA | 33.5 ± 6.37 | 3.06 ± 1.2 | 10.25 ± 3.37 | NA | 1960 ± 650 | 1790 ± 550 |
| Devasahayam et al., 2020 | Canada | A 10-week (3x/week) Bodyweight supported treadmill (BWST) (aerobic training) | 40 min BWST, including 5 min of warm-up and cool-down. | BDNF, IL-6 | ELISA (R&D Systems, Inc., Minneapolis, MN, USA) Serum | 10 (3 PPMS, 7 SPMS) | NA | 53.2 ± 15.6 | NA | NA | 9/1 | 67620 ± 20430 | 63460 ± 19970 |
| Banitalebi et al., 2020 | Iran | A 12-week (3x/week) (Combined training) | (IG1, IG2, IG3) 1° Resistance exercise: 3 sets of 12 reps at 40–70% RM. 2° Aerobic exercise: Cycling or running. 20 min at 50–70% HR$_{max}$ 3° Balance training 4° Stretching exercises | BDNF, GDNF, CNTF, NT-3, NT-4 | ELISA Serum | 45 (RRMS) | NA | 40.83 ± 8.17 | NA | NA | NA | 1822700 ± 925700 | 2552506.7 ± 1121929.5 |
| Devasahayam et al., 2021 | Canada | Graded exercise training (GXT) (aerobic training) | 80 steps per minute during GXT and the workload was increased in ~20-watt increments every 2 min, starting from load level 3 (21 watts) until exhaustion | BDNF, IL-6 | ELISA (R&D Systems, Inc., Minneapolis, MN, USA) Serum | 14 (3 PPMS, 11 SPMS) | McDonald criteria | 54.07 ± 8.46 | NA | NA | 10/4 | 56560 ± 25120 | 56470 ± 44110 |
| Savšek et al., 2021 | Slovenia | A 12-weeks (2x/week) aerobics (aerobic training) | 60 min, consisting of a 6–10 min warm-up, 30–40 min performed at prescribed intensity, and a 6–10 min cool-down | BDNF, IL-6 | ELISA (R&D Systems, Inc., Minneapolis, MN, USA) Serum | 14 (RRMS) | NA | 39.7 ± 6.7 | 2.94 ± 1.61 | 8.4 ± 6.1 | 11/3 | 1923.52 ± 1260.9 | 2011.94 ± 672.48 |

Abbreviations: EDSS: The Expanded Disability Status Scale, BDNF: Brain-derived neurotrophic factor, IL-6: Interleukin-6, IL-10: Interleukin-10, sIL-6R: Soluble IL-6, NGF: Nerve growth factor, TNF-α: Tumor necrosis factor α, SOCS: Suppressor of cytokine signaling, MMP: Matrix metalloproteinase, S1P: Sphingosine-1-phosphate, IGF: Insulin-like Growth Factor, GDNF: Glial cell line-derived neurotrophic factor, CNTF: Ciliary neurotrophic factor, NT: Neurotrophin, RRMS: Relapsing-remitting multiple sclerosis, SPMS: Secondary-progressive multiple sclerosis, PPMS: Primary-progressive multiple sclerosis, VDBP: Vitamin D-binding protein.

**Table 2. Result of quality assessment of the included studies in the meta-analysis according to the PEDro scale.**

| Author, Year | Schulz et al., 2004 | Bansi et al., 2013 | Wens et al., 2016 | Ozkul et al., 2018 | Eftekhari et al., 2018 | Khademosharie et al., 2018 | Zimmer et al., 2018 | Jørgensen et al., 2019 | Abbaspour et al., 2020 | Devasahayam et al., 2020 | Banitalebi et al., 2020 | Devasahayam et al., 2021 | Savšek et al., 2021 |
|---|---|---|---|---|---|---|---|---|---|---|---|---|---|
| **PEDro Items** | | | | | | | | | | | | | |
| Eligibility criteria | | * | * | | * | * | * | * | * | * | * | * | * |
| Random allocation | * | * | * | * | * | | * | * | * | | * | | * |
| Concealed allocation | | * | | | | | * | * | | | * | | |
| Baseline comparability | * | * | * | * | * | | * | * | * | * | * | * | * |
| Masked participants | | | | | | | | | | | | | |
| Masked therapists | | | | | | | | | | | | | |
| Masked assessors | | * | | * | | | * | | * | | * | | * |
| Adequate follow-up | * | * | * | * | * | * | * | * | | * | * | * | * |
| Intention to treat analysis | | | | | | | * | | | | | | * |
| Between-group statistical comparison | * | * | * | * | * | * | * | * | * | * | * | * | * |
| Point estimates and variability | * | * | * | * | * | * | * | * | * | * | | * | * |
| *score* | 5 | 8 | 6 | 6 | 6 | 4 | 9 | 7 | 6 | 5 | 7 | 5 | 8 |

Astriks* means Yes. Blank means No.

0.0161, Fig 2). A drapery plot is shown to visualize the meta-analysis results based on *p-value* functions of each study (*p-value* on the y-axis and the effect size on the x-axis) (S1 Fig).

The Egger's test (*p-value* = 0.38) (S2 Fig), the Begg's test (*p-value* = 0.36), and funnel plots (S3 Fig) showed no evidence of publication bias. The Eggers' test does not indicate the presence of substantial funnel plot asymmetry. A Q value of 24.74 and an $I^2$ index of 51.5% indicate significant heterogeneity and inconsistency, respectively, among included studies.

Using the 'find.outliers' command in R software, the study by Zimmer et al. [42] was detected to be an outlier; thus, we repeated the meta-analysis and the result is as follows: (SMD 0.26, 95% CI [0.05; 0.47], *p-value* = 0.0189, test of heterogeneity: $I^2$ = 6.4%, Q = 11.75, *p-value* = 0.4, Fig 3).

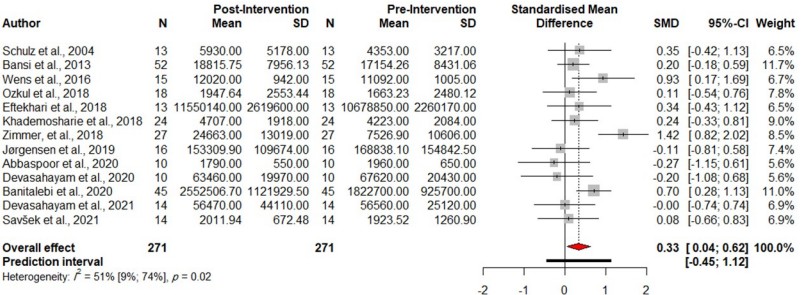

**Fig 2. Forest plot of meta-analysis of pre- and post-intervention levels of BDNF.**

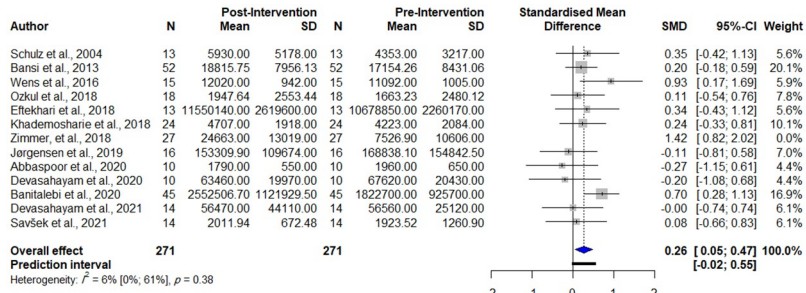

**Fig 3. Forest plot of meta-analysis of pre- and post-intervention levels of BDNF removing outliers.**

Sensitivity analysis (leave-one-out analysis) showed that the effect size remained significant after omitting each study, and the heterogeneity did significantly reduce (S4–S6 Figs).

Additionally, to find the sources of heterogeneity, metaregression was conducted. The heterogeneity between studies could partially be explained by the PEDro score ($R^2$ = 15.32%). No correlation was found between the effect size and the mean age of the EDSS score. The results of metaregression analysis are shown as bubble plots (Fig 4).

### 3.6. Aerobic exercise vs. combined exercise

Subgroup analysis revealed no significant difference between aerobic and combined exercise subgroups (Q = 0.03, *p-value* = 0.86) (Fig 5).

### 3.7. Duration of exercise intervention

We performed a subgroup analysis to assess the effect of the duration of physical exercise on BDNF levels. Out of 12 studies that reported the duration of their intervention, five scheduled the program for 12 weeks or more, and seven conducted the program for less than 12 weeks. The subgroup analysis did not show a significant difference between the two abovementioned groups (Between groups Q = 0.07, *p-value* = 0.7914, Fig 6). The results for each group are as follows: Less than 12 weeks: (n = 7, SMD 0.3193, 95%CI [-0.2039; 0.8425] Q = 16.67, $I^2$ = 64.0%), 12 weeks or more: (n = 5, SMD 0.3941, 95%CI [-0.1200; 0.9083], Q = 6.86, $I^2$ = 41.7%).

## 4. Discussion

The main finding of our study is that physical exercise significantly increases baseline serum levels of BDNF in PwMS. Subgroup analysis did not reveal any significant difference due to the type of exercise and the duration of the exercise training program.

Our meta-regression suggested that the existing heterogeneity in the meta-analysis results was not significantly related to sex or chronological age. It is plausible that variations in the quality of included studies could partially explain the existed heterogeneity in the meta-analysis results. Variations in the BDNF measurement procedures and techniques (as pointed out in the study by Hirsch et al. [49]), protocol and settings of physical activity programs, and the ratio of included participants with RRMS or progressive MS in the included investigations might be other contributors to the heterogeneity in the meta-analysis results. We were not able to evaluate the association between exercise program intensity and session time and changes in baseline peripheral BDNF concentrations. As all studies reported serum concentrations of BDNF, it is plausible that resting peripheral BDNF concentration is not a source of heterogeneity in this analysis.

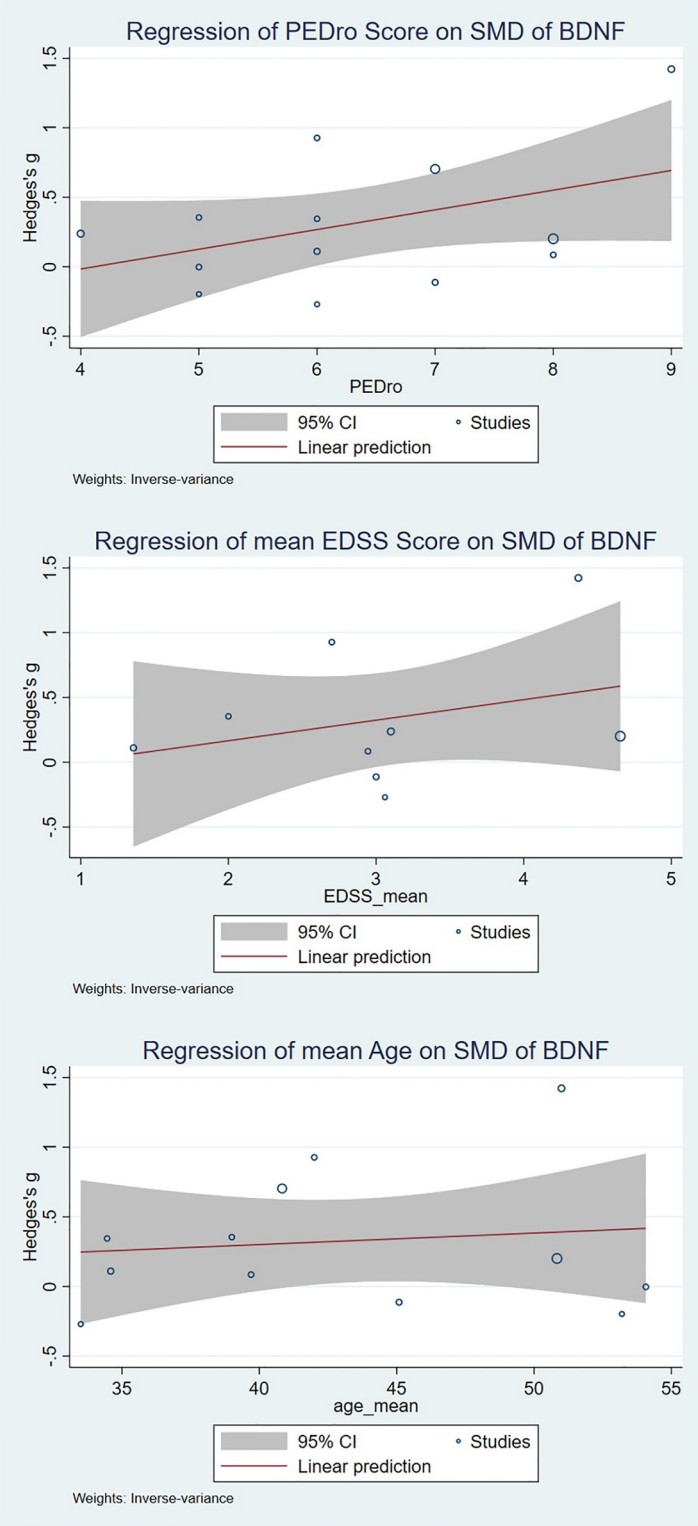

**Fig 4. Results of meta-regression.** Effects of age, PEDro score, and mean expanded disability status scale (EDSS) score on the effect size of the comparisons of pre-and post-intervention levels of BDNF were assessed wherever for 10 or more original studies data was available.

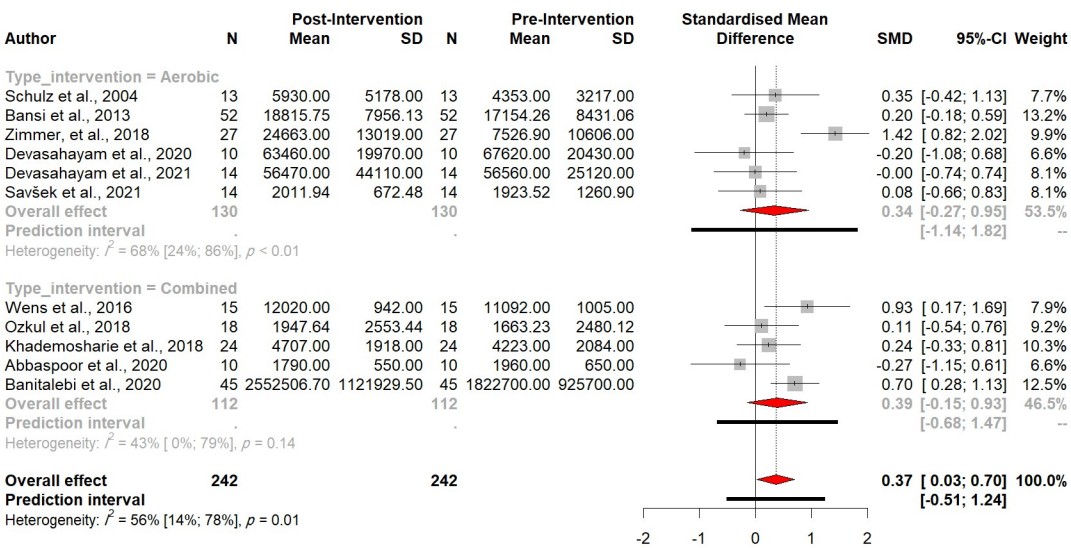

**Fig 5. Forest plot of the subgroup analysis (type of intervention).**

Several pieces of evidence suggest that physical activity contributes to CNS functioning through multiple mechanisms [50–52], including (1) increasing cerebral blood flow, (2) endocannabinoid and neurotransmitter modulation, (3) alterations in neuroendocrine responses, and (4) structural changes in the CNS [50–52]. Given that exercise increases BDNF levels, this has been regarded as one of the potential mechanisms by which exercise affects brain health and functioning. BDNF improves synaptic potentiation, synaptic plasticity, and neuronal activity, also modifying axodendritic morphology [7,53,54]. In addition, BDNF enhances quantal neurotransmitter release by influencing presynaptic terminals, which potentiates synaptic transmission [55].

Exercise-induced increase of BDNF levels can contribute to plastic changes following physical activity [7,56]. Although no previous study has simultaneously investigated BDNF and neuroimaging changes after exercise in PwMS, several studies have demonstrated the effect of exercise in brain structure. Prakash et al. [57] used a cross-sectional design to demonstrate that

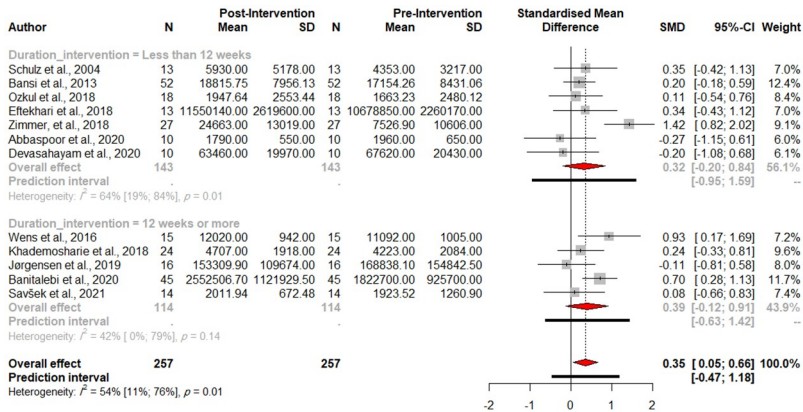

**Fig 6. Forest plot of the subgroup analysis (duration of intervention).**

aerobic exercise affects both grey matter volume and white matter integrity in PwMS. Klaren et al. [58] reported that the volume of several areas of the brain, including the hippocampus, thalamus, caudate, putamen, and pallidum, relates to the level of moderate/vigorous physical activity. Kjølhede et al. [59] showed that the trend of reduced brain atrophy is nonsignificant in PwMS who attended a six months (2 days/week) resistance training program.

Multiple central and peripheral factors, influence the levels of BDNF [10]. In MS, treatment [60,61], sex [62], age [63], body mass index (BMI) [64], and disease status, as assessed by EDSS [65,66], have been associated with the peripheral levels of BDNF. Our results confirm that physical activity can also influence BDNF levels in PwMS. As the increase in serum levels of BDNF may reflect both peripheral and CNS changes [2], it remains to be determined the main source and/or target of these enhanced BDNF levels.

A previous meta-analysis of studies measuring BDNF after exercise in healthy adults revealed that aerobic training can increase BDNF concentrations more than resistance training [12]. Moreover, the duration of exercise was highly associated with BDNF increase [10]. In contrast, our subgroup meta-analysis considering the type of intervention (aerobic vs. resistance training) and duration of exercise (less than 12 weeks vs. 12 weeks or more) showed no significant difference in test for between-groups differences (random-effects model).

Frequency, intensity, duration, and mode of exercise are essential variables in the context of achieving positive rehabilitation outcomes for adults with MS [67,68]. Our subgroup meta-analysis considering the type of intervention (aerobic vs. resistance training) and duration of exercise (less than 12 weeks vs. 12 weeks or more) showed no significant difference in test for between-groups differences (random-effects model). Recent interest in High Intensity Interval Training suggests that alternative exercise modalities may also offer promise in inducing BDNF related neuroplasticity in PwMS [69,70].

Collectively, reported data on the benefits of physical therapy and exercise on cognitive functions, including memory, learning, information processing, attention, and concentration, is noteworthy [71–75]. Numerous research have been conducted to date on the function of BDNF in the acute exercise—cognition connection [76–82]. Although all these studies corroborate the positive effects of acute exercise on cognitive performance, discrepancies exist in the proof that BDNF is responsible for these cognitive advantages [83]. Considering the previously-mentioned studies, exercise-induced alterations in BDNF and its correlation with cognitive performance were specifically tested by Ferris et al. [76], Lee et al. [79], Skriver et al. [80], Tsai et al. [82], and Winter et al. [78]. Additionally, substantial correlations between acute exercise-induced modifications in BDNF concentration and cognitive function have been reported in research measuring memory, but not in studies investigating non-memory aspects of cognition [83]. Regarding the beneficial effects of aerobic exercise, a case study by Leavitt et al. [84] reported a 16.5% increase in hippocampal volume following a 53% improvement in their memory after a 12-week aerobic exercise (30 min, 3 days/week). A systematic review addressing studies on MS and cognition exhibited that out of eight included studies, five studies showed that exercise positively impacts patients' cognitive status [8]. Additionally, BDNF Val66Met polymorphism has shown protective roles against cognitive impairment in PwMS [85]. Moreover, evidence indicates the neurotrophic role of BDNF on motor-related neurons, which may relieve motor symptoms via modulating neuronal morphology and motility [86].

There is some evidence suggesting that exercise may reduce MS progression. Regarding this association, intense exercise has shown reductions in MS development, excluding known factors and determinants of MS progression [87]. Still, long-term follow-up studies are needed to confirm these findings. In addition, based on pieces of evidence given in a systematic review, physical exercise training may reduce the risk of relapse in PwMS by 27% in the training group versus the control group [9]. However, a lack of papers with standard methodologies

assessing the association of exercise training and MS progression exists in the current literature.

Notably, for other neurodegenerative disorders, including Parkinson's disease (PD) and Alzheimer's disease (AD), exercise-induced BDNF has been proposed as a protective factor. For example, a meta-analysis of two randomized controlled trials and four pre-experimental studies with a total of 100 patients with PD undergoing physical exercise showed a significant increase in BDNF blood levels in parallel with improvement of motor symptoms (e.g., improvement in Unified Parkinson's disease rating scale-Part III (MDS-UPDRS-III)) [49]. Another meta-analysis reporting the effects of exercise on neurotrophic factors in cognitively impaired individuals diagnosed with dementia or mild cognitive impairment (MCI) demonstrated positive and significant effects of exercise resulting in higher inflammatory and neurotrophic biomarkers in MCI patients [88].

Future studies should study the association between exercise, neuroplasticity markers and functional outcomes to determine whether the observed neurotrophic effects translate to clinical benefits. Ultimately, based on research findings, exercise may be an invaluable adjunct component to the existing medical treatments. Of note, more human studies are needed to underpin this relationship, as we cannot only rely on animal studies because of humans and animals' structural and functional brain differences.

The main limitation of this current meta-analysis is that the extant studies on exercise in PwMS involved relatively small number of subjects. Other limitations include short-term interventions (<26 weeks), low levels of disability of most participants (EDSS scores <4), and the inclusion of mainly relapsing-remitting MS or mixed-group patient populations and exclusion of patients with comorbidities that could their relief assessed due to exercise training. Moreover, we only included the intervention group of the included studies and analyzed the pre- and post-intervention levels of BDNF in PwMS who underwent exercise, as another meta-analysis conducted by Ruiz-González et al. [89] compared post-intervention levels of BDNF in both PwMS and controls. It is worth emphasizing that Mackay et al. [90] intended to assess the impact of aerobic exercise on BDNF levels in persons with neurological disorders without segregating neurologic conditions (e.g., MS). Although they performed the meta-analysis considering all of the neurologic disorders, their results align with the current study and indicate that exercise might contribute to increased BDNF concentrations.

## 5. Conclusion

This systematic review and meta-analysis show that physical activity increases peripheral levels of BDNF in PwMS. Future studies involving more subjects across different forms of the disease undergoing well-designed physical interventions are warranted. These studies should also incorporate multiple measures of BDNF alongside neuroimaging outcomes. These initiatives will ultimately strengthen the quality and scope of the evidence on MS rehabilitation [20].

## Supporting information

**S1 Checklist. PRISMA 2020 checklist.**
(DOCX)

**S1 Fig. Drapery plot of meta-analysis of pre- and post-intervention levels of BDNF.**
(DOCX)

**S2 Fig. Egger's plots of the meta-analysis.**
(DOCX)

**S3 Fig. Funnel plot and counter-enhanced funnel plot of meta-analysis of BDNF.**
(DOCX)

**S4 Fig. Results of sensitivity analysis (leave-one-out analysis) of the meta-analysis (Baujat plot).**
(DOCX)

**S5 Fig. Results of sensitivity analysis (leave-one-out analysis) of the meta-analysis (influence diagnostics).**
(DOCX)

**S6 Fig. Results of sensitivity analysis (leave-one-out analysis) of the meta-analysis (I2 and effect size plot).**
(DOCX)

**S1 Appendix. Search strategies, and funnel, drapery, Egger's, and sensitivity analysis plots.**
(DOCX)

# Acknowledgments

We sincerely thank the authors of the included articles in this review due to sharing the relevant data.

# Author Contributions

**Conceptualization:** Parnian Shobeiri.

**Data curation:** Parnian Shobeiri, Amirali Karimi, Sara Momtazmanesh.

**Formal analysis:** Parnian Shobeiri.

**Investigation:** Parnian Shobeiri, Amirali Karimi, Sara Momtazmanesh.

**Methodology:** Parnian Shobeiri.

**Supervision:** Nima Rezaei.

**Visualization:** Parnian Shobeiri.

**Writing – original draft:** Parnian Shobeiri.

**Writing – review & editing:** Amirali Karimi, Sara Momtazmanesh, Antônio L. Teixeira, Charlotte E. Teunissen, Erwin E. H. van Wegen, Mark A. Hirsch, Mir Saeed Yekaninejad, Nima Rezaei.

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
