## [Decision Letter · Decision Letter 0]

7 Oct 2021

PONE-D-21-27156Exercise-induced increase in blood-based brain-derived neurotrophic factor (BDNF) in people with multiple sclerosis: a systematic review and meta-analysis of exercise intervention trialsPLOS ONE

Dear Dr. Nima Rezaei,

Thank you for submitting your manuscript to PLOS ONE. After careful consideration, we feel that it has merit but does not fully meet PLOS ONE’s publication criteria as it currently stands. Therefore, we invite you to submit a revised version of the manuscript that addresses the points raised during the review process.

We look forward to receiving your revised manuscript.

Kind regards,

Cosme F. Buzzachera, Ph.D.

Academic Editor

PLOS ONE

Journal Requirements:

"Erwin E.H. van Wegen and Mark A. Hirsch were funded by The Dutch Brain Foundation."

We note that you received funding from a commercial source: The Dutch Brain Foundation

Reviewers' comments:

Reviewer's Responses to Questions

**Comments to the Author**

1. Is the manuscript technically sound, and do the data support the conclusions?

Reviewer #1: Yes

Reviewer #2: Yes

2. Has the statistical analysis been performed appropriately and rigorously? 

Reviewer #1: Yes

Reviewer #2: Yes

3. Have the authors made all data underlying the findings in their manuscript fully available?

Reviewer #1: Yes

Reviewer #2: Yes

4. Is the manuscript presented in an intelligible fashion and written in standard English?

Reviewer #1: Yes

Reviewer #2: Yes

5. Review Comments to the Author

Reviewer #1: This is a systematic review and meta-analysis of exercise intervention trials focused on the exercise-induced serum levels of brain-derived neurotrophic factor (BDNF) in persons with multiple sclerosis (pwMS).

The review is very comprehensive, the analysis is well conducted, and deals with a particularly relevant topic in the field of MS. In particular, it confirms that concentrations of serum BDNF increase after exercise, supporting the hypothesis that physical activity is useful for promoting neuroprotection through brain plastic changes, which in turn is particularly relevant in limiting MS progression.

We must take into account the limitations that affect most of the studies analyzed, which make the conclusions refer to relatively selected patient cohorts (low levels of neurological disability, mainly relapsing-remitting courses , exclusion of patients with comorbidities).

In addition, the reported findings are not particularly original, since they are also reported by other meta-analyzes. In particular, the current study is quite similar to a previous one (The Effect of Aerobic Exercise on Brain-Derived Neurotrophic Factor in People with Neurological Disorders: A Systematic Review and Meta-Analysis. Mackay CP, Kuys SS, Brauer SG. Neural Plast. 2017), which also highlighted the effect of exercise on BDNF levels in patients with neurological diseases (MS included), and that should be discussed.

Reviewer #2: Title:

Exercise-induced increase in blood-based brain-derived neurotrophic factor (BDNF) in people with multiple sclerosis: a systematic review and meta-analysis of exercise intervention trials.

General comments:

The authors are commended for a well-written manuscript. In my opinion, the investigation is well-structured, and all methodological standards for this kind of study were observed. I believe there are only minor issues with the manuscript in its current form that need to be addressed before publication. I hope that the following critique will be received in the way it is delivered and be used to improve the quality of the manuscript.

Minor concerns:

The following minor concerns are presented in order of appearance.

Abstract.

The authors stated that meta-analyses have not been previously performed comparing pre- and post-intervention BDNF concentrations in patients with multiple sclerosis, however, in the discussion section (lines 388-389) they cited the meta-analysis by Ruiz-Gonzàles et al. (2021). Please check these statements and if appropriate reformulate the sentences. In addition, did the authors check the articles collected in the Ruiz-Gonzàles’ review for their work?

Introduction.

The Introduction is well-structured, however, there is no information about mechanisms lying to the increase of BDNF due to exercise. Why exercise increases the BDNF level? If possible, could you provide a brief explanation about this?

Results. Lines 198-200.

In this sentence, the authors reported the studies’ countries. This information is also present in table 1. In my opinion, this information is not relevant to the aim of the study, so it is not necessary to report it also in the text. Please consider removing this sentence.

Results. Lines 270; 331; and Fig. 9.

Regarding the subgroup analysis, the authors refer to two kinds of exercise: aerobic vs anaerobic, but sometimes they referred to anaerobic exercise as resistance training. However, in table 1, only one article was classified as resistance training, while another one that used pilates as training was not considered in any classification but reported as an anaerobic exercise during the analysis. Can “Anaerobic exercise” and “Resistance training” terms be considered interchangeably? The subgroup analysis was performed considering also combined training, but it was not reported in the title. Please consider specify better the classification criteria and reporting the same terminology in all parts of the manuscript. Consider also re-grouping the articles in table 1 following the final classification.

Discussion. Lines 323-328; 333-336.

These two paragraphs report the same information. Please consider removing the second.

Discussion. Lines 339-343.

In my opinion, this paragraph should be re-written to enhance comprehension of the link between the benefits of exercise on cognitive function and the role of the BDNF.

Discussion. Lines 356-357.

This sentence seems to be unrelated to the previous information, while the quality of life of people with MS depends largely on the MS progression. Please consider expanding information about exercise, MS progression and quality of life in multiple sclerosis or remove the sentence.

Discussion. Lines 363.

“improvement in” is in bold.

Discussion. Lines 368-374.

Please consider moving this information into the introduction section.

Figures 1-10.

The number of figures in the manuscript is high. Please consider including only relevant ones and move the rest (e.g., 3, 4, 5, and 7) to the supplementary.

6. PLOS authors have the option to publish the peer review history of their article (what does this mean?). If published, this will include your full peer review and any attached files.

Reviewer #1: No

Reviewer #2: **Yes: **Luca Correale

---

## [Author Response · Author response to Decision Letter 0]

11 Dec 2021

Dear Prof. Buzzachera,

Thank you for your consideration of the manuscript "Exercise-induced increase in blood-based brain-derived neurotrophic factor (BDNF) in people with multiple sclerosis: a systematic review and meta-analysis of exercise intervention trials" by PLoS One.

Our responses to the reviewers' queries are discussed below. A revision-marked draft of the revised manuscript is respectfully enclosed accordingly to the instructions.

In case any other revision might be required, please kindly let us know. We would do our bests to make them at the earliest.

Thank you for your consideration of this manuscript. 

Sincerely,

Nima Rezaei, MD, PhD

Mir Saeed Yekaninejad, PhD

Journal Requirements:

A1: Many thanks. The manuscript is in line with PLOS ONE’s requirements. 

"Erwin E.H. van Wegen and Mark A. Hirsch were funded by The Dutch Brain Foundation."

We note that you received funding from a commercial source: The Dutch Brain Foundation

A2: Dear Editor,

Many thanks for considering our manuscript. We should declare that we did not receive any funding from The Dutch Brain Foundation which is a non-for-profit organization. Also, Erwin E.H. van Wegen and Mark A. Hirsch declared that they did not receive any financial fundings (e.g., salary, research funding) from The Dutch Brain Foundation. So, we removed the previous statement in the competing interests section and declared “Not applicable.” As we mentioned before in our request for an APC fee waiver during the submission process, we are from a low-middle-income country, and unfortunately, we cannot afford the APC fee. However, we are sure that this article would have a significant impact and several benefits for PLOS ONE, as several articles might cite it in the future. Necessary to mention that, this does not alter our adherence to PLOS ONE policies on sharing data and materials.

A3: We revised the manuscript and we added a sort of references to the reference list. We tried to provide complete and correct references. Please contact us in case of any inconveniences. 

Editor’s Comments to Author:

1. Please upload a copy of Figure 7-10 which you refer to in your manuscript. Or if the figure is no longer to be included as part of the submission please remove all reference to it within the text.

A1: Dear Editor,

Many thanks for considering our manuscript. In the revised manuscript with track changes, there is no references for figures 7-10. Only 6 figures are cited in the revised manuscript. The previous version had 10 cited figures, so that we move 4 figures to the supplementary material.

2. Please ensure that you refer to SI Figures 3-6 in your text as, if accepted, production will need this reference to link the reader to the figure.

A2: We sincerely thank you for considering our manuscript. SI figures 3-6 are cited in the revised manuscript, on page 16, lines 259 and 269, respectively.

3. Based on the information you've provided, we've drafted for your approval the following proposals to update your funding statements:

- Financial Disclosure

"The authors received no specific funding for this work."

- Competing Interests

"The authors have no competing interests to declare."

Please confirm whether the above proposals are accurate, and if so, whether we may update your statements accordingly on your behalf

A3: Many thanks for your comments to our manuscript. Yes, we confirm the above-mentioned competing interests and financial disclosure statements. Also, we have updated the manuscript regarding these issues in line 442 and 443, on page 24.

Reviewer(s)' Comments to Author:

Reviewer: 1

Comments to the Author

Q1: This is a systematic review and meta-analysis of exercise intervention trials focused on the exercise-induced serum levels of brain-derived neurotrophic factor (BDNF) in persons with multiple sclerosis (pwMS).

The review is very comprehensive, the analysis is well conducted, and deals with a particularly relevant topic in the field of MS. In particular, it confirms that concentrations of serum BDNF increase after exercise, supporting the hypothesis that physical activity is useful for promoting neuroprotection through brain plastic changes, which in turn is particularly relevant in limiting MS progression.

We must take into account the limitations that affect most of the studies analyzed, which make the conclusions refer to relatively selected patient cohorts (low levels of neurological disability, mainly relapsing-remitting courses, exclusion of patients with comorbidities).

In addition, the reported findings are not particularly original, since they are also reported by other meta-analyzes. In particular, the current study is quite similar to a previous one (The Effect of Aerobic Exercise on Brain-Derived Neurotrophic Factor in People with Neurological Disorders: A Systematic Review and Meta-Analysis. Mackay CP, Kuys SS, Brauer SG. Neural Plast. 2017), which also highlighted the effect of exercise on BDNF levels in patients with neurological diseases (MS included), and that should be discussed.

A1: Thank you for your great comment. We are happy for your valuable comment and consideration. We discussed the study by Mackay et al. on page 22, lines 398-402. Changes are highlighted accordingly. Notably, the study by Mackay et al. analyzed RCT/CT studies measuring BDNF after exercise training. Though, they did not divide the included studies as separate groups due to their included patients’ conditions. In the current meta-analysis, we only included MS patients. And this highlights the specificity of our manuscript. Moreover, the study by Mackay et al. confirms our findings regarding the effect of exercise on increasing BDNF levels. 

Reviewer: 2

Comments to the Author

Q1: General comments:

The authors are commended for a well-written manuscript. In my opinion, the investigation is well-structured, and all methodological standards for this kind of study were observed. I believe there are only minor issues with the manuscript in its current form that need to be addressed before publication. I hope that the following critique will be received in the way it is delivered and be used to improve the quality of the manuscript.

Minor concerns:

The following minor concerns are presented in order of appearance.

A1: Many thanks for your valuable feedback and consideration. 

Q2: Abstract.

The authors stated that meta-analyses have not been previously performed comparing pre- and post-intervention BDNF concentrations in patients with multiple sclerosis, however, in the discussion section (lines 388-389) they cited the meta-analysis by Ruiz-Gonzàles et al. (2021). Please check these statements and if appropriate reformulate the sentences. In addition, did the authors check the articles collected in the Ruiz-Gonzàles’ review for their work?

A2: Thank you for your great comment. Ruiz-Gonzàles et al. investigated the changed of BDNF due to exercise in two different groups (MS vs. Control). However, we analyzed the pre- and post-intervention BDNF concentrations in a single group of people with multiple sclerosis. Ours differs to that cited in the discussion in this way that we included different groups in the meta-analysis. Regarding your second question, yes, we checked the articles collected in the Ruiz-Gonzàles’ review to make sure of including all the relevant studies in the literature. 

Q3: Introduction.

The Introduction is well-structured, however, there is no information about mechanisms lying to the increase of BDNF due to exercise. Why exercise increases the BDNF level? If possible, could you provide a brief explanation about this?

A3: Thank you for your helpful comment. We provided information regarding the mechanisms that exercise affects BDNF concentrations on page 5, lines 99-104. Changes are highlighted accordingly. 

Q4: Results. Lines 198-200.

In this sentence, the authors reported the studies’ countries. This information is also present in table 1. In my opinion, this information is not relevant to the aim of the study, so it is not necessary to report it also in the text. Please consider removing this sentence.

A4: Thank you for your comment. We removed the sentences that you have mentioned. Changes are tracked and highlighted accordingly. 

Q5: Results. Lines 270; 331; and Fig. 9.

Regarding the subgroup analysis, the authors refer to two kinds of exercise: aerobic vs anaerobic, but sometimes they referred to anaerobic exercise as resistance training. However, in table 1, only one article was classified as resistance training, while another one that used pilates as training was not considered in any classification but reported as an anaerobic exercise during the analysis. Can “Anaerobic exercise” and “Resistance training” terms be considered interchangeably? The subgroup analysis was performed considering also combined training, but it was not reported in the title. Please consider specify better the classification criteria and reporting the same terminology in all parts of the manuscript. Consider also re-grouping the articles in table 1 following the final classification.

A5: Many thanks for your precise comment. We reconsidered the sub-group analysis. The most correct way is to make 2 main sub-groups, Aerobic and Combined training. We removed anaerobic training sub-group, as it was not reasonably correct, and therefore Eftekhari et al. and Jørgensen et al. were removed from the sub-group analysis. Fig5 was updated and relevant changed were highlighted in the manuscript accordingly. Terms used in the table 1 are the exact terms used by the authors to describe their type of exercise. The title is updated to “Aerobic Exercise vs. Combined Exercise.”

Q6: Discussion. Lines 323-328; 333-336.

These two paragraphs report the same information. Please consider removing the second.

A6: Thank you for your comment, which helps improve our manuscript. We removed the second paragraph that you have mentioned. Changes are tracked and highlighted accordingly.

Q7: Discussion. Lines 339-343.

In my opinion, this paragraph should be re-written to enhance comprehension of the link between the benefits of exercise on cognitive function and the role of the BDNF.

A7: Thank you for your comment. We added additional information on page 20 in lines 348-357, regarding the current literature on the role of BDNF in the acute exercise—cognition relationship. 

Q8: Discussion. Lines 356-357.

This sentence seems to be unrelated to the previous information, while the quality of life of people with MS depends largely on the MS progression. Please consider expanding information about exercise, MS progression and quality of life in multiple sclerosis or remove the sentence.

A8: Thanks for your comments and consideration. We removed the sentence that you have mentioned. Changes are tracked and highlighted accordingly.

Q9: Discussion. Lines 363.

“improvement in” is in bold.

A9: Thank you for your comment. We addressed your comment, and it is highlighted accordingly. 

Q10: Discussion. Lines 368-374.

Please consider moving this information into the introduction section.

A10: Many thanks for your comment. We put this information on page 5-6, Lines 105-111. Changes are highlighted accordingly. 

Q11: Figures 1-10.

The number of figures in the manuscript is high. Please consider including only relevant ones and move the rest (e.g., 3, 4, 5, and 7) to the supplementary.

A11: Many thanks for your comments. We rearranged the figures and updated our supplementary material. Changes are highlighted in the manuscript, accordingly. Additionally, the legends of figures and Supporting information captions were updated and highlighted.

---

## [Decision Letter · Decision Letter 1]

14 Feb 2022

Exercise-induced increase in blood-based brain-derived neurotrophic factor (BDNF) in people with multiple sclerosis: a systematic review and meta-analysis of exercise intervention trials

PONE-D-21-27156R1

Dear Dr. Nima Rezaei,

We’re pleased to inform you that your manuscript has been judged scientifically suitable for publication and will be formally accepted for publication once it meets all outstanding technical requirements.

Kind regards,

Cosme F. Buzzachera, Ph.D.

Academic Editor

PLOS ONE

Reviewers' comments:

Reviewer's Responses to Questions

**Comments to the Author**

1. If the authors have adequately addressed your comments raised in a previous round of review and you feel that this manuscript is now acceptable for publication, you may indicate that here to bypass the “Comments to the Author” section, enter your conflict of interest statement in the “Confidential to Editor” section, and submit your "Accept" recommendation.

Reviewer #2: All comments have been addressed

2. Is the manuscript technically sound, and do the data support the conclusions?

Reviewer #2: (No Response)

3. Has the statistical analysis been performed appropriately and rigorously? 

Reviewer #2: (No Response)

4. Have the authors made all data underlying the findings in their manuscript fully available?

Reviewer #2: (No Response)

5. Is the manuscript presented in an intelligible fashion and written in standard English?

Reviewer #2: (No Response)

6. Review Comments to the Author

Reviewer #2: (No Response)

7. PLOS authors have the option to publish the peer review history of their article (what does this mean?). If published, this will include your full peer review and any attached files.

Reviewer #2: **Yes: **Luca Correale

---

## [Editor Report · Acceptance letter]

23 Feb 2022

PONE-D-21-27156R1 

Exercise-induced increase in blood-based brain-derived neurotrophic factor (BDNF) in people with multiple sclerosis: a systematic review and meta-analysis of exercise intervention trials 

Dear Dr. Rezaei:

I'm pleased to inform you that your manuscript has been deemed suitable for publication in PLOS ONE. Congratulations! Your manuscript is now with our production department. 

Kind regards, 

on behalf of

Dr. Cosme F. Buzzachera 

Academic Editor

PLOS ONE